# Conformational Landscape of Cytochrome P450 Reductase Interactions

**DOI:** 10.3390/ijms22031023

**Published:** 2021-01-20

**Authors:** Manuel Sellner, André Fischer, Charleen G. Don, Martin Smieško

**Affiliations:** Computational Pharmacy, Departement of Pharmaceutical Sciences, University of Basel, 4056 Basel, Switzerland; manuel.sellner@unibas.ch (M.S.); and.fischer@unibas.ch (A.F.); charleen.don@unibas.ch (C.G.D.)

**Keywords:** Cytochrome P450 reductase, CPR, conformational, dynamics

## Abstract

Oxidative reactions catalyzed by Cytochrome P450 enzymes (CYPs), which constitute the most relevant group of drug-metabolizing enzymes, are enabled by their redox partner Cytochrome P450 reductase (CPR). Both proteins are anchored to the membrane of the endoplasmic reticulum and the CPR undergoes a conformational change in order to interact with the respective CYP and transfer electrons. Here, we conducted over 22 microseconds of molecular dynamics (MD) simulations in combination with protein–protein docking to investigate the conformational changes necessary for the formation of the CPR–CYP complex. While some structural features of the CPR and the CPR–CYP2D6 complex that we highlighted confirmed previous observations, our simulations revealed additional mechanisms for the conformational transition of the CPR. Unbiased simulations exposed a movement of the whole protein relative to the membrane, potentially to facilitate interactions with its diverse set of redox partners. Further, we present a structural mechanism for the susceptibility of the CPR to different redox states based on the flip of a glycine residue disrupting the local interaction network that maintains inter-domain proximity. Simulations of the CPR–CYP2D6 complex pointed toward an additional interaction surface of the FAD domain and the proximal side of CYP2D6. Altogether, this study provides novel structural insight into the mechanism of CPR–CYP interactions and underlying conformational changes, improving our understanding of this complex machinery relevant for drug metabolism.

## 1. Introduction

Cytochrome P450 enzymes (CYPs) are the most relevant superfamily of drug-metabolizing enzymes occurring in humans, animals, plants, and microorganisms. Besides tasks in intrinsic metabolic processes, they are responsible for the detoxification of xenobiotics within organisms by the chemical modification of substrates for downstream conjugation and excretion [1,2]. CYP2D6 is responsible for the metabolic transformation of approximately 25% of marketed drugs [3] and is subject to a high degree of polymorphism, leading to diverse interindividual metabolic profiles [4]. In order to perform the wide spectrum of oxidative reactions, CYPs depend on a redox partner, the Cytochrome P450 reductase (CPR), transferring electrons from the cofactor nicotinamide adenine dinucleotide phosphate (NADPH) to the heme iron atom of the CYP. The CPR consists of the flavin mononucleotide (FMN) domain and the flavin adenine dinucleotide (FAD) domain, which bind the corresponding cofactors and transfer electrons from NADPH via FAD and FMN [5,6]. Further, the FAD and FMN domains are connected by a linker domain containing a flexible hinge region between itself and the FMN domain (Figure 1A) [6,7]. The interaction of the CPR and the CYP as well as the subsequent electron transfer (ET) require a conformational transition of the CPR from the closed to the open state, which remains incompletely understood in its details [6,8]. In the aforementioned conformational conversion, it is thought that the FMN domain undergoes a rotational or stretching motion caused by adaptations of the hinge region [6,9,10]. Further, the redox state of the cofactors and the salt concentration of the surrounding environment were proposed to be involved in the conformational transition [7,9,11]. As the CPR is the main redox partner serving electrons to all microsomal CYPs [6], conformational adaptations to fine-tune these individual interactions are necessary.

The drug-metabolizing CYPs as well as the CPR are anchored to the membrane of the endoplasmic reticulum in eukaryotic cells [1,6,12]. The functionality of both CYPs and the CPR is known to be influenced by the microenvironment of the membrane. For example, it was shown that the anchoring to the membrane positively influences the thermodynamics of the FAD reduction by NADPH [13]. Further, protein–membrane interactions of CYP2D6 have been shown to influence its structure [14]. Thus, the structural model of the full-sequence protein including the alpha-helical membrane anchor is necessary for a complete understanding of CPR–CYP interactions and underlying conformational adaptations. As the active site of CYPs is buried within the core of the enzyme, so-called tunnels connect it to the surrounding solvent environment and influence the substrate specificity of the enzyme. While tunnels responsible for the uptake of mostly lipophilic substrates of CYPs face the membrane, the ones that are thought to be relevant for the release of hydrophilic molecules such as oxidized substrates or water molecules protrude into the solvent [3,14,15,16,17]. The membrane attachment of both CYP2D6 and the CPR complicates crystallization procedures as the anchors have to be omitted to solubilize the proteins [18]. On the other hand, computational methods, such as structural modeling coupled to molecular dynamics (MD) simulations, allow the investigation of conformational changes in atomistic detail by providing insight into time-evolved structural dynamics [19]. In previous work, different types of restraints were introduced in the simulations to obtain the open conformation of the CPR [6]. However, this study was limited by a comparatively short simulation time. In CYP3A4, MD simulations revealed that the interaction with the FMN domain of the CPR induced the opening of a tunnel, which was proposed to be involved in the egress of water molecules to desolvate the active site and facilitate ligand binding [20]. Further, the interactions of CYP2B4 and the CPR were characterized using site-directed mutagenesis experiments and showed the importance of positively charged residues of the CPR involved in protein–protein interactions [21].

Here, we applied an integrated computational protocol consisting of protein–protein docking as well as conventional MD simulations and metadynamics simulations to highlight conformational changes necessary for the interaction of CYP2D6 and the CPR, residues relevant for the protein–protein interaction, as well as the influence of the CPR on the structure of CYP2D6. Besides confirming previous structural observations, we revealed a mechanism for conformational changes induced by the alteration of FMN oxidation state, which is based on the flipping of a glycine residue coupled to a change in the local interaction network. Further, we present novel insight into the conformational transitions required for CPR–CYP interactions by characterizing an upright position of the CPR relative to the membrane. Remarkably, our simulations of the CPR–CYP2D6 complex highlighted a previously unknown interaction surface for stabilizing protein–protein contacts. Lastly, we detected a widening of the access tunnels to the buried active site of CYP2D6 upon interaction with the CPR. In conclusion, we provide novel structural insight into the mechanism of CPR–CYP interactions and underlying conformational changes, improving our understanding of this complex machinery relevant for drug metabolism.

## 2. Results and Discussion

### 2.1. Open Conformation of the CPR

As mentioned above, the conformational transition of the CPR from a closed state, which favors intramolecular electron transport, to an open state, capable of interacting with its redox partners such as CYPs, is still insufficiently understood [6,8]. As the membrane environment is relevant for the functionality of both proteins [3,13,14,16], we constructed a full-length model of the membrane-anchored CPR. Using this model, we aimed to simulate a transition of the CPR from a closed conformation to an open conformation by conventional MD simulations. Details on the construction of the model are given in the Materials and Methods section. In brief, all systems simulated with conventional MD in this study presented sufficient to optimal convergence based on the computed root mean square deviation (RMSD) values, indicating stability of the helical architecture (Appendix A). The radius of gyration can be used to characterize the conformational state of the CPR and was experimentally determined to amount to 27 Å in the closed conformation [6]. In accordance with the above-mentioned experiments, we observed average values ranging from approximately 26 to 2 Å in our simulations.

For the description of the redox states of the cofactors, we use the subscripts _ox_, _sq_, and _hq_ to denote the oxidized, semiquinone, and hydroquinone state, respectively. At first, we conducted simulations of the FAD_ox_/FMN_hq_ redox state with a duration of 1.25 μs in triplicate (Appendix A). Experiments have shown an influence of the ionic strength on the metabolic rates of reactions supported by the CPR, potentially due to the strong electrostatic interactions involved in maintaining inter-domain contact between FMN and FAD [9,22]. Therefore, we exposed the CPR to a salt concentration of 0.5 M NaCl, as it was proposed to maximize its reactivity, in three 0.48 μs simulations. In addition, the mutation of Arg243 in the hinge region of the CPR to alanine was proposed to shift its structural ensemble toward the open conformation due to the removal of a positive charge and the consequent loss of ionic interactions between the hinge and the linker domain [9,12]. Thus, we conducted another triplicate of 1.44 μs conventional simulations to investigate the effects of the R243A mutation. As indicated by the radius of gyration, the distance between the FMN and FAD cofactors (Appendix A), as well as visual inspection of the trajectories, conventional simulations did not present the conformational transition from the closed to the open state independent of the surrounding or structural adaptations. This suggests the closed conformation to be stable beyond the microsecond timescale, which is in accordance with experiments suggesting a timescale of 55 s^−1^ for the interflavin ET [23]. Hence, the complete conformational transition likely takes place beyond the timescale of our simulations. Further, a recent study implied the closed conformation to be energetically favored [11], rendering it unlikely to observe a transition in the microsecond timescale despite our efforts promoting it.

To overcome the sampling limitations inherent to conventional MD, we conducted metadynamics simulations to investigate the opening of the CPR. In this protocol, we selected the distance between the centroids of the FAD and FMN domain (denoted as CV1) as well as the distance between the centroid of the linker domain and residues located in proximity to the FMN cofactor (denoted as CV2) as collective variables. In addition to plain metadynamics (Appendix A), we conducted well-tempered metadynamics simulations to benefit from an adaptive height of the Gaussian kernels. This provided better defined local minima in the energy landscape of the CPR (Figure 1D). Our model of the open conformation obtained by metadynamics simulations presented a similar domain orientation to the structure of the open rat CPR, with only minor differences in the FMN domain (Figure 1B,C). The FMN cofactor was in approximately the same location in both structures, while the N-terminal part of the FMN domain in the open CPR obtained by metadynamics was slightly rotated away from the membrane. This rotation was facilitated by the high flexibility of the hinge connecting the FMN domain to the linker domain. The observed conformational differences might have been caused by the lack of a redox partner, as the rat open CPR was crystallized in complex with heme oxygenase. Variations in the amino acid sequences may have also contributed to the observed structural differences, as the sequence identity between the two structures amounts to 93.9% (Appendix A). The radius of gyration in our open CPR increased to 28.6 Å, which is close to the 28.0 Å in the rat open CPR. Moreover, the interflavin distance increased from 4 to 26.5 Å (compared to 26.1 Å in the rat open CPR), indicating a reasonable opening of the domains.

In the obtained free energy profiles, the open conformation of the CPR did not present a clear energetic minimum (Figure 1D). This stands in accordance with previous observations, in which high stability of the closed conformation was observed in multiple computational studies [11,22]. On the other hand, the closed conformation presented two distinct local energetic minima in regard to the selected collective variables. The first minimum occurred with CV1 at approximately 33.5 Å and CV2 at 27 Å, indicating a reasonable closed conformation, which could be confirmed by visual inspection and superposition to crystal structures. The second minimum occurred with CV1 at approximately 34 Å and CV2 at 35 Å. This conformation presented a slight rotation of the FMN domain towards the membrane, leading to an increased interflavin distance of 15.7 Å (Appendix A).

#### 2.1.1. Transitions between a Sitting and Upright Position

As mentioned above, the CPR serves electrons to a variety of proteins. It was previously shown that its interaction partners such as CYPs present different isotype-specific insertion depths of the globular domain in the membrane (how deeply the CYP is immersed in the membrane) [17]. Hence, in order to favorably interact with the conserved region on the distal side of drug-metabolizing CYPs [21,22,24], a mechanism to raise its globular domain to different heights relative to the membrane would be convenient for the CPR to structurally adapt to different interaction partners. Indeed, we observed a rising motion of the CPR in respect to the membrane, independent of any redox partner, in several unbiased MD simulations (Figure 1E). The whole globular domain of the enzyme reproducibly transitioned from a sitting to an upright position (Appendix A). During this transition, most of the superficial membrane contacts were detached and the transmembrane anchor adopted a more perpendicular orientation relative to the membrane plane. As the enzyme retained a closed conformation during this motion, our results are in accordance with the Förster resonance energy transfer (FRET) experiments, suggesting a closed conformation in the absence of a redox partner [25]. The described rising motion started with the detachment of the FAD domain from the membrane, which occurred at a distance between the CPR and the membrane of around 48 Å. While this effect was independent of the redox state of the CPR, it was most pronounced in the simulations with increased ionic strength and mutation of R243 (Appendix A). To the best of our knowledge, such a conversion from a sitting to an upright position independent of a redox partner has not been previously described. However, it is worth noting that another study suggested a change of the insertion depth of CYP1A1 in complex with the CPR [24]. Potentially, both a change in the insertion depth of the CYP and a rising motion of the CPR could occur dependent on the combination of the CPR with different redox partners.

#### 2.1.2. Glycine Flip upon Reduction of FMN

Computational studies and site-directed mutagenesis experiments revealed inter-domain contacts between the FAD and FMN domain to be mediated by salt bridges [6,26]. In flavodoxins, which are ET proteins in bacteria [27], a glycine residue was shown to interact with the FMN cofactor. In particular, the oxidized form of the flavin accepts a hydrogen bond from the backbone amide of glycine. Upon reduction, the corresponding nitrogen atom of FMN is protonated, leading to a conformational change in the glycine residue, which reorients from an O-down to an O-up position. Due to this reorientation, the glycine backbone oxygen can accept a hydrogen bond from the reduced cofactor (Figure 2A,B) [28,29]. In the CPR, this glycine residue is conserved (Gly141) and its deletion abolishes the catalytic activity of the enzyme due to the missing stabilization of the FMN semiquinone state [30]. The FMN domain of the CPR aligns to the flavodoxin of *Clostridium beijnerinckii* (PDB ID: 1FLA) with a backbone RMSD of 2.9 Å, underlining their similarity (Figure 2C). Thus, we monitored the conformation of Gly141 in our simulations to validate whether a similar mechanism could occur in the human CPR. Indeed, we observed the described transition from O-down to O-up in multiple simulations (Appendix A). As the validity of this mechanism is in accordance with available crystal structures, our results underline the analogy of the CPR to the related flavodoxins in regard to this flip. It was further proposed that the glycine flip disrupts the surrounding interaction network, leading to the opening of the reductase by inducing the reorientation of Glu142 involved in inter-domain interactions between FAD and FMN [7]. However, we rarely observed such a transition of Glu142 in our unbiased simulations and, when it occurred, it was not associated with a separation of the domains.

### 2.2. Interactions between CYP2D6 and the CPR

To obtain insight into the interaction of the CPR with CYP2D6, we conducted five replica simulations of the unliganded CPR–CYP2D6 complex and recorded the individual protein RMSD values as well as the distance between the FMN and heme cofactors (Figure 3). The RMSD of the globular domain of the CPR ranged from 4.6 to 7.9 Å, whereas the RMSD of CYP2D6 was significantly lower, with values below 2 Å, suggesting the structural adaptations that facilitate the protein–protein interaction to be mostly on the side of the CPR. The root mean square fluctuation (RMSF) of the CPR presented maxima around the linker region (residues 210–250), as well as a loop region with intrinsic flexibility (Figure 3). The latter was not resolved in the crystal structure selected for this project, indicating the high flexibility of this loop. This is in accordance with site-directed mutagenesis experiments suggesting the flexibility of the hinge region to be essential for a potent interaction with different redox partners [12,31]. The interaction surface of the FMN domain and CYP_BM3_, which is a bacterial fusion protein of a CYP and an electron-donating FMN domain, is known from crystallography [32]. This allowed us to compare our model complex since reasonable conservation of the interactions among CYP isoforms was proposed [31,33]. The average distance between the heme and FMN cofactors ranged from 15.4 to 17.8 Å (Figure 3) and, thus, was slightly shorter than in the crystal structure of CYP_BM3_, where the distance amounts to 18.3 Å [22,32]. However, one replica simulation presented a larger distance of 21.7 Å. A recent study on the CPR–CYP1A1 complex similarly reported distances shorter than in CYP_BM3_ [24]. In addition to simulations with unliganded CYP2D6, we conducted simulations with the three substrates tramadol, paracetamol, and promethazine [34] bound to the CYP2D6 active site. While it has been reported that substrate binding to CYP2D6 reduces the distance between FMN and heme from around 18 Å to less than 15 Å [5], we could not observe such an effect in a total of 4.5 μs of simulation time (Appendix A). In CPR–CYP complexes, the interflavin distance was determined to be around 36 Å in the open conformation based on NMR studies [35]. All our simulations presented values in accordance with these experiments, with distances between 27 and 40 Å. The capability of the CPR–CYP complex for ET is strongly dependent on the distance between the cofactors, as well as the distance between the heme moiety of the CYP and the FMN domain [24,36]. Based on the computed distances in our CPR model, interflavin ET would only be possible in the closed conformations, whereas an ET from FMN to heme would be possible over the average measured distances of 16 to 19 Å when compared to 18.3 Å in CYP_BM3_.

Next, we superimposed our CPR–CYP2D6 complexes to CYP_BM3_ and could determine a reasonable overlap at an RMSD of 4.7 Å (Figure 4A). Remarkably, we could identify two distinct interaction sites between the CPR and CYP2D6. The first interface involved three negatively charged residues of the CPR (Glu142, Asp144, and Glu179) located in loops around FMN. Their counterparts were two arginine residues in CYP2D6 in proximity to the heme (Arg140, Arg440), as shown in Figure 4B and Appendix A. This first interaction interface has been reported in previous studies [6,21]. While the above-mentioned interactions occurred in all complexes for at least 75% of the simulations, we additionally detected Arg132 and Arg133 of CYP2D6 to be involved in inter-protein interactions in 60% of the complex systems. These two arginine residues similarly interact with the negatively charged patch surrounding the FMN cofactor of the CPR and were proposed to help with conveying electrons from FMN to the D-proprionate of the heme [37]. At the second interface, Asp337 in the J’ loop of CYP2D6 interacted with multiple positively charged residues of the FAD domain (Figure 4B). In the existing literature, we could not find any reference to this second interaction site involving CYPs [10]. However, the same interface is present in the crystallized complex of rat CPR and heme oxygenase [38]. Potentially, this second interaction surface further stabilizes the ET complex by favorable protein–protein contacts and reduction of the solvent-exposed area of both proteins. Such an extended interaction surface is in accordance with the strong affinity of the CPR–CYP interaction, which was measured to be in the low nanomolar range [24]. Hence, our results confirmed the known importance of electrostatic complementarity between the two proteins [12] and suggest an additional recognition interface between the two proteins that was not previously identified in drug-metabolizing CYPs.

### 2.3. CPR Binding Affects Tunnels in CYP2D6

A large amount of evidence points toward the presence of tunnels involved in the uptake of ligands to the buried active site of drug-metabolizing CYPs [3,14,15,16,17,20,20]. In previous work, it was proposed that the interaction of CYP3A4 with the CPR induces the opening of the so-called water tunnel (tunnel W, nomenclature according to Cojocaru et al. [39]). This tunnel is thought to be involved in the egress of water molecules from the active site upon the association of ligands [20]. Further, two very recent studies highlighted the opening of access tunnels of CYPs upon association of the CPR. While Mukherjee and colleagues could not detect significant changes in their conventional MD simulations, they observed alternative egress tunnels in their random acceleration MD simulations with CYP1A1 in the presence of the CPR [24]. The latter protocol applies a randomly directed force on the ligand molecule, which is adapted if the ligand does not cover a certain distance to explore egress pathways [40]. In contrast to an altered distribution of tunnels, the second study proposed facilitated uptake of substrates into the active site of human aromatase upon the interaction with the CPR based on the widening of tunnel 2b [33]. As the need for additional studies focused on the impact of the CPR on substrate access to CYPs was discussed [24], we investigated this in our CPR–CYP2D6 complexes.

Using the obtained trajectories, we analyzed the impact of CPR–CYP interactions on the bottleneck radii of tunnels in CYP2D6. The bottleneck radius describes the width at the most narrow point along a tunnel, where gating residues frequently control the passage of ligands. While crystal structures only provide a static view of the protein, MD simulations allow the analysis of time-evolved changes in the protein and, thus, also the dynamic ligand tunnels within it [15,41]. Besides bottleneck radii, another metric that can be used to characterize tunnels is their opening time along a trajectory [16]. Similar to the work on CYP1A1 and human aromatase mentioned above, we detected increased bottleneck radii as well as higher opening times of tunnels in the CPR–CYP2D6 complex as opposed to the uncomplexed CYP (Figure 5A and Appendix A). In contrast, we could not detect any changes in tunnel W, even though it was suggested to open upon complex formation [20]. Tunnel 2b, located among the BC loop, FG loop, and the β1 sheets in CYP2D6 (Figure 5B), was proposed to be one of the most relevant tunnels for substrate uptake [3,16]. While tunnel 2b showed increased bottleneck radii in the CPR–CYP2D6 complex, the differences were not statistically significant at the *p* = 0.05 level. On the other hand, tunnel 2f, which is located among the FG loop, the β4 sheets, and the N-terminus of helix A in proximity to tunnel 2b, presented a statistically significant (at *p* = 0.05) opening in all complexes with the exception of the one with acetaminophen bound to CYP2D6. In the latter triplicate, one simulation presented a comparatively high opening time, leading to an increased standard deviation and, therefore, a lack of significance. In our previous work, we observed tunnel 2f to be relevant for the uptake of acetaminophen from the protein–membrane interface [3]. Interestingly, the presence of ligands bound to CYP2D6 increased the opening time of tunnel 2f even more. Hence, our results further substantiate the impact of CPR interactions on access tunnels by facilitating the passage of ligands to the buried active sites of CYPs.

## 3. Materials and Methods

### 3.1. Simulation Parameters

If not specified otherwise, we used the following parameters for all conventional MD simulations presented in this work. We selected the Desmond (v2016-4) simulations engine [42] coupled to the OPLS_2005 force field. All complexes were placed in orthorhombic periodic boundary systems with TIP3P solvent molecules as well as an appropriate amount of sodium or chloride ions to neutralize the systems. The u-series algorithm [43] was used to treat long-range forces, with a cutoff of 9 Å for short-range interactions. Bonds to hydrogen atoms were constrained using the M-SHAKE algorithm. The simulations were conducted in an NPT ensemble at a temperature of 310.15 K and atmospheric pressure controlled by the Nose–Hoover thermostat (relaxation time 1.0 ps) and the Martyna–Tobias–Klein barostat (relaxation time 2.0 ps), respectively. Prior to the production phase, for which we retained a timestep of 2.0 fs for the RESPA integrator, each system was equilibrated using the default workflow in Desmond.

### 3.2. Model Building

For the globular part of the CPR, we used a crystal structure of the human wild-type CPR (PDB ID: 5FA6). We used the Protein Preparation Wizard [44] included in the Maestro Small-Molecule Drug Discovery Suite (v2018-2) [45] to add hydrogen atoms, assign bond orders, predict protonation states at pH 7.4, and reorient the hydrogen bonding network with PROPKA at pH 7.4. Further, the missing residues (numbers 504–509) in a flexible loop of the FAD domain were automatically added by the Prime protocol. Next, the structure was refined by a restrained heavy atom minimization using the OPLS_2005 force field toward an RMSD convergence threshold of 0.3 Å. Since NADPH was not fully resolved in the selected crystal structure, we exchanged it with NADPH from another crystal structure (PDB ID: 3ES9) after superposition of the proteins (Appendix A). To mimic its physiological environment, we inserted the protein into a membrane consisting of a prequilibrated 1-palmitoyl-2-oleoylphosphatidylcholine (POPC) bilayer. The position of the protein relative to the membrane was predicted using the Positioning of Proteins in Membrane (PPM) server and imported into the System Builder panel in Maestro [46]. To equilibrate the system, we conducted a 30 ns MD simulation to obtain a representative structure clustering using the trajectory_cluster.py routine included Maestro (v2016-4). To obtain a full atomistic model of the human CPR, we modeled the membrane anchor as an ideal alpha helix in Maestro based on the sequence in the UniProt [47] database (UniProt ID: P16435). The positioning of the anchor was similarly predicted using the PPM server and we inserted it so that residues 22–43 were embedded in the membrane. The embedded anchor was equilibrated for 200 ns and, to obtain a representative structure, the trajectory was clustered as described above. In the next step, both representative structures were merged through a covalent bond while retaining a hydration shell of 6 Å around the protein and enlarging the membrane. Next, this full-length system was equilibrated for 200 ns and clustered using the method described above. A representative structure of the most populated cluster was used to model the different redox states. For one system, we added a fully solved structure of NADP^+^ from a rat open CPR (PDB ID: 3ES9) by superimposing it with the partially solved nicotinamide moiety in our structure (Appendix A). Since the Maestro did not allow the modeling of free radicals required for the semiquinone states of FAD and FMN, we manually adjusted the partial charges according to Appendix A.

To investigate the complete electron transfer complex, we selected CYP2D6 as the model enzyme due to its importance in drug metabolism in approximately 25% of marketed drugs, its high degree of genetic polymorphism, and our prior experience with this protein [3]. To obtain the CPR–CYP2D6 complex, we preprocessed the CYP2D6 crystal structure (PDB ID: 3TDA) as described above. As it is known that the CPR binds to CYPs in an open conformation [6], we applied a flexible multidomain docking protocol using the HADDOCK (v2.2) online server [48] and the previously prepared structure of the CPR. We separated the CPR at its hinge region between residues 232 and 233 and restrained their distance to 1.4 Å, with lower and upper margins of 0.1 and 8.6 Å, respectively. The residues between numbers 232 and 243 were specified to be treated with full flexibility, while semi-flexible handling of of the remaining residues was enabled. Further, we defined a center of mass constraint with a force constant of 1.0 and all residues with a solvent accessibility of at least 40% were defined as active residues in the docking protocol (Appendix A). We selected the CPR–CYP2D6 complex with the lowest RMSD to the crystal structure of the rat CPR and heme oxygenase (PDB ID: 3WKT) as our final structure and connected the CPR back together at the cleavage site. The residues around this region (231–243) were minimized using the 3D Builder in Maestro with restraints on the remaining protein before adding the membrane anchor as described above. In this ET complex, we selected the fully reduced redox state of FMN and fully oxidized FAD. Since the output from HADDOCK did not include non-polar hydrogen atoms, we processed the structure with the Protein Preparation Wizard as described above. Next, we replaced the globular part of the CYP with a full-length membrane-anchored model described in our previous work [16]. In a last step, we created a new system including both proteins in their full-length embedded in a membrane, resulting in a total of 207,789 atoms. In addition to the simulations with unliganded CYP2D6, we decided to build additional systems with different substrates bound to the CYP2D6 active site. A complex of CYP2D6 with acetaminophen was retained from our previous work [3]. Next, a starting structure of CYP2D6 bound to promethazine was modeled according to a crystal structure bound to thioridazine (PDB ID: 4WNW), in which we manually modified the ligand and superimposed the complex to our model of CYP2D6. As the crystal structure provided two different orientations of thioridazine, we selected the one with the methyl sulfide closer to the heme. To obtain a structure bound to tramadol, we manually adapted a pose obtained from docking using the smina protocol [49]. The docking pose was modified in order to position the methoxy group in an orientation which would allow O-demethylation [50] based on the distance of the site of metabolism to the heme iron [51]. During the modification, we avoided steric clashes and obtained a pose with a heavy atom RMSD of 1.8 Å to the best scored pose from docking.

### 3.3. Conventional MD Simulations and Post-Processing

Using the above described model of the membrane-anchored CPR, we modeled seven different redox states (Appendix A). However, we decided to set our primary focus on the fully reduced state of FMN with oxidized FAD. Simulations with the R243A mutation were conducted in triplicate for 1440 ns, while those with increased salt concentrations were conducted for 480 ns in triplicate. For the complex of full-length CPR and CYP2D6 with both enzymes anchored to the membrane, we conducted five replicas with a duration of 300 ns. In addition, the above-mentioned ligand–protein complexes of acetaminophen, promethazine, and tramadol were individually simulated with five replicas for a duration of 300 ns. Further, to obtain a baseline, we conducted five replica simulations of CYP2D6 in the absence of the CPR for 300 ns.

The sequence alignment between human and rat CPR was computed using the ClustalW algorithm within the UGENE [52] toolkit (v1.32.0) based on FASTA sequences derived from the UniProt IDs P16435 and P00388, respectively. The sequences were truncated to represent the entries in the PDB files and visualized using JalView (v2.10.5) [53]. The identity was determined based on mismatches between the sequences. To compute the tunnels within CYP2D6, we used the CAVER (v3.01) program, which allows the processing of MD trajectories [41]. To determine the absolute starting point for the tunnel computation, the CAVER Analyst (v2.0) program [54] was used by selecting Glu216, Asp301, and the heme of CYP2D6 as active site residues. Using a probe radius of 0.9 Å and a clustering threshold between 4.0 and 5.0 Å (Appendix A), we analyzed every second frame of all simulations of the CPR–CYP2D6 complexes. To determine the significance of differences in the bottleneck radii and opening times, we performed a one-tailed Welch *t*-test. Statistical significance was determined at a *p*-value of 0.05. The raising motion of the CPR was quantitatively shown by measuring the distance between the centers of mass of the membrane and the globular part of the CPR (residues 67–677). To quantify the flip of Gly141, we measured the psi and phi angles in all frames of the simulations. The Simulation Interaction Diagram and the Simulation Event Analysis tool included in Maestro were used to calculate the RMSD, RMSF, radius of gyration, and distances between atoms in all frames of the conducted simulations. We used the C8 methyl carbon atoms of FAD and FMN to calculate the interflavin distance, and the distance between FMN and heme was defined as the distance between the C7 methyl carbon atom of FMN and the heme Fe atom.

### 3.4. Metadynamics Simulations

Metadynamics simulations were conducted using the Desmond simulation engine. We defined two collective variables: CV1 as the distance between the centers of mass of the FAD domain and the FMN domain and CV2 as the distance between the center of mass of the linker domain and the center of mass of residues 88, 141, 178, and 210. These individual residues are located in the FMN domain, particularly in loops surrounding the FMN cofactor. An increase in CV1 would lead to a stretching motion of the CPR, whereas CV2 would allow the rotation of the FMN domain. Both mechanisms were previously described to be involved in the conformational transition between open and closed [10]. To keep the CVs from increasing to an unnatural extent, we set a wall at 50 and 45 Å for CV1 and CV2, respectively. For both CVs, a Gaussian width of 0.2 Å was defined. The metadynamics simulations were conducted with a Gaussian height of 0.03 kcal/mol and a deposition interval of 0.09 ps. Regarding the redox states, we modeled fully oxidized FAD and fully reduced FMN as cofactors, while retaining the simulation parameters from conventional MD. In addition, we conducted a well-tempered metadynamics simulation with a kTemp factor of 10 for 240 ns.

## 4. Conclusions

To efficiently catalyze the metabolism of xenobiotics, CYPs depend on the CPR to donate electrons during their catalytic cycle. Their interaction is based on the formation of an ET complex, for which the CPR has to undergo a transition from a closed to an open conformation. Here, we investigated this transition following protein–protein interactions with an atomistic model of the full-length membrane-anchored CPR, as well as the CPR–CYP2D6 complex with an integrated modeling protocol consisting of protein–protein docking, conventional MD, as well as metadynamics simulations. While we could not detect a conformational transition from a closed to an open state in our conventional simulations, we reproducibly observed a motion during which the CPR transitioned from a sitting to an upright position relative to the membrane independent of a redox partner. As the CPR serves a diverse set of enzymes, including CYPs with different burying depths within the membrane, such a mechanism could benefit the resulting protein–protein interactions. Further, we could confirm a mechanism that was observed in prokaryotic flavodoxins, in which a glycine residue in close proximity to the FMN cofactor flips depending on the present redox state. This flip is thought to alter the inter-domain interaction network and promote an open conformation. We further investigated the influence of an increased salt concentration in the medium as well as the mutation of Arg243, as these perturbations were suggested to induce the conformational transition. However, in the studied timescale, we could not observe large reorientations of the protein in these modified conditions. Due to the inherent limitation of conventional MD simulations to study rare conformational changes, we conducted metadynamics simulations which introduce bias towards specific collective variables, for which we selected two intramolecular distances within the CPR. In these simulations, we could detect the opening of the CPR, which was mostly mediated by a stretching motion of the FMN domain, accompanied by a slight rotation. The flexibility of the hinge region connecting the FMN and the linker domain was essential for this rearrangement. We validated the open conformation of the CPR with an available crystal structure of the rat CPR in an open conformation, which has a sequence identity of 93.9% to the human enzyme, and found a considerable overlap with our proposed structure. Similarly, our CPR–CYP2D6 complex, selected from the protein–protein docking, presented high identity to the crystal structure of CYP_BM3_, which is a fusion protein consisting of a domain similar to human CYPs coupled to an FMN domain. Interestingly, our full-length model displayed an additional interaction interface between CYP2D6 and the FAD domain of the CPR, which was not previously described in drug-metabolizing CYPs. By comparing simulations with uncomplexed CYP2D6 and CPR–CYP2D6 complexes bound to different ligands, we could confirm the previously proposed alteration of ligand access tunnels by this protein–protein interaction. Our results advance the understanding of this complex machinery, which is highly relevant in the metabolism of drugs and other xenobiotics.

## Figures and Tables

**Figure 1 ijms-22-01023-f001:**
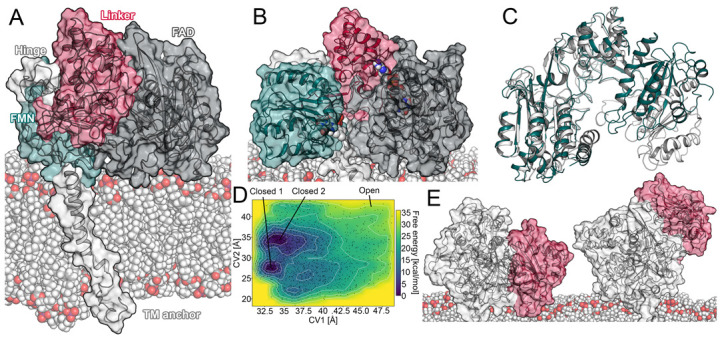
Conformational changes in the CPR. (**A**) Domains of the CPR. (**B**) Open conformation of the CPR obtained from well-tempered metadynamics simulation. Domain coloring was retained from (**A**). (**C**) Superposition of the aforementioned open conformation of the CPR and the open conformation of the rat CPR (PDB ID: 3WKT). (**D**) Free energy profile of well-tempered metadynamics simulation. (**E**) Sitting and upright position of the CPR (the FAD domain is colored red).

**Figure 2 ijms-22-01023-f002:**
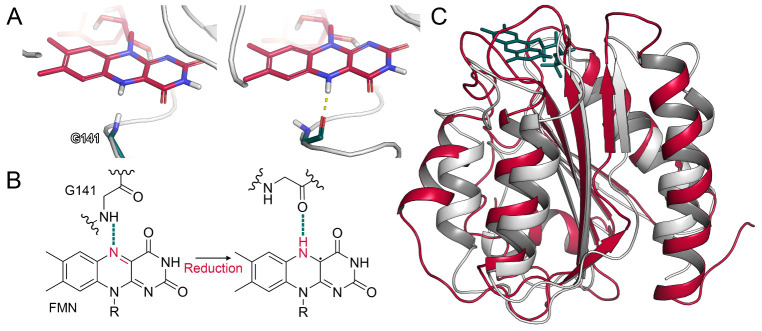
Conformational changes in the CPR and CPR–CYP2D6 complex. (**A**) Glycine flip in a simulation of FAD_ox_/FMN_hq_. (**B**) Schematic representation of glycine flip. (**C**) Superposition of CPR FMN domain and a flavodoxin (PDB ID: 1FLA).

**Figure 3 ijms-22-01023-f003:**
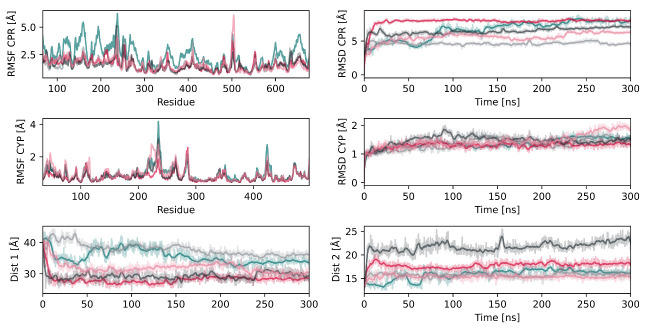
Metrics of the CPR–CYP2D6 unliganded complex simulations conducted with five replicas. The RMSD and RMSF of each protein are presented along the distance between FAD and FMN (denoted as Dist 1) and between the heme and FMN (denoted as Dist 2).

**Figure 4 ijms-22-01023-f004:**
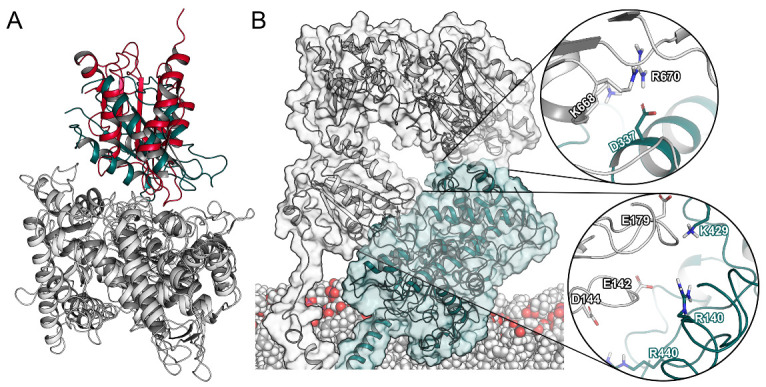
Protein–protein interactions in the CPR–CYP2D6 complex. (**A**) Superposition of CPR–CYP2D6 complex and CYP_BM3_ (PDB ID: 1BVY). (**B**) Complex of the CPR and CYP2D6 with spotlight on the interaction surfaces.

**Figure 5 ijms-22-01023-f005:**
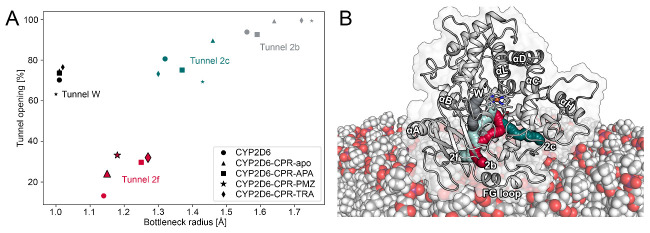
Results of tunnel computation. (**A**) The average bottleneck radii and relative opening times of nine different tunnels in CYP2D6. Symbols with increased size and with an outline indicate that the represented opening time differs significantly (at a significance level of 0.05) from the corresponding value of free CYP2D6. (**B**) Depiction of tunnels in CYP2D6.

## Data Availability

The data presented in this study are openly available in our GitHub repository at https://github.com/mmodbasel/CPR_conformation/.

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
