# Peer review of "Conformational Landscape of Cytochrome P450 Reductase Interactions"

_ijms, 2021, doi:10.3390/ijms22031023_

Round 1

Reviewer 1 Report

Very good manuscript with only some minor issues to be addressed.

  • The rationale for choosing CYP2D6 among all available CYP isoforms should be provided. Why not CYP3A4 as it is the most important isoform for drug metabolism?
  • The role of Arg 243 and other key residues should be explained more fully
  • Generally, more figures (such as superimposed structures mentioned at several places throughout the manuscript) should be provided to make the manuscript more readable and intuitive.
  • Some vague terms are used throughout manuscript, that should be explained more fully, e.g.: l203 “reasonable overlap, l307 “the most similar”, l315 “appropriately sized membrane”, etc. I suggest checking the whole manuscript and trying to avoid this kind of expression.
  • Validation of methods used in this study should be briefly mentioned within the main manuscript.

Author Response

Please, find the answers with additional graphics in the attached PDF document.

Reviewer 2 Report

  Structural basis of CPR-CYP complex is crucial to understand ET reaction between them, however, it is still uncertain. Authors carried out MD simulations of the full-length CPR which includes the membrane-anchor region and the CPR-CYP2D6 complex. Although the conventional MD simulation approach dost not reproduce the structural transition of the open and closed of CPR as well as the previous reports, the metadynamics approach is possible to reproduce the open form of CPR structure. During these approaches, authors also detected the positional switching of CPR relative to the membrane which may contribute to interact with several redox partners. The complexed model of CPR-CYP2D6 suggests that CYP2D6 interacts with both FMN and FAD domain of CPR and the opening of the tunnels to the active site of CYP2D6 is induced by the interaction with CPR. The results sound nice and appropriate for publication in International Journal of Molecular Science with major revisions as shown below.

Major comments

  • In page 3, line 113-120, authors compared the human open CPR structure simulated by authors with rat open CPR structure determined by crystallography. Authors detected the minor difference in the orientation of FMN domain and estimated that it is caused by the sequence difference. However, rat open CPR structure determined by crystallography is a part of solubilized CPR in the complex structure of CPR and heme oxygenase whereas the simulated human CPR structure is full-length CPR which includes anchor region. Further NADP is bound to rat open CPR whereas NADP is unbound in the simulated human open CPR structure. Thus, the structural difference may be caused by the binding of heme oxygenase or NADP. NADP-unbound rat open CPR structure in complex with heme oxygenase has been already reported (Sugishima M et al (2019) FEBS lett. DOI: 10.1002/1873-3468.13360). Comparison with NADP-unbound rat open CPR structure may be useful to eliminate the effect of NADP induced conformation change of CPR.
  • In page 4, line 140-141, authors discussed about the transition of CPR from a sitting to an upraised position relative to membrane. What are the representative distances between CPR and membrane for a sitting and an upraised positions? If the 52 Å distance is the criteria to discriminate two positions, the position switching is only observed in FADox/FMNhq state (Figure S11) and it may depend on the redox states of CPR although authors are not mentioned it.
  • In page4, line 147, authors described “conformational transition”. In my understanding, it means transitions between a sitting and upraised position and does not mean the conformational transition between the closed and open conformation. This phrase is confusing for readers.
  • In the section 2.2, authors compared the human open CPR structure in complex with CYP2D6 with the crystal structure of CYPBM3. As described in line 184, CYPBM3 is a bacterial fusion protein of P450 domain (heme-containing domain) and the domain corresponding with CPR. Authors used the word CYPBM3 or P450BM3 ambiguously. This word sometimes means P450 domain of CYPBM3, sometimes means full-length of CYPBM3. Please clarify it.
  • In page 6, line 188, authors used the word “CPR-CYPBM3 complex”, but the crystal structure is the complex of FMN domain and P450 domain of CYPBM3. FAD domain and linker domain were not contained in the structure.
  • In line 212-214, authors noted the second interface between CPR and CYP2D6 and claimed that this was not previously described. It may be new in the interaction of CPR with CYP but this second interface is reported in the crystal structure of the complex of rat CPR and heme oxygenase (Sugishima M et al (2014) Proc. Natl. Acad. Sci. USA DOI: 1073/pnas.1322034111). In this structure, the positively charged interface of the FAD domain interacts with the negatively charged “G-helix” of heme oxygenase (Fig.6 in that paper). The interaction of CPR with heme oxygenase must be noted in this paper and accordingly referred.

Minor comments

  • Abbreviations (ox, sq, hq) must be denoted somewhere.
  • Reference 23 is insufficient.
  • The sentence shown in page 3, line84 must be revised because experimental parameters are described in Materials and Methods section and not in the Supplementary Information.
  • The contents represented in Figure 2 are the mixture of the contents explained in the section 2.1.2 (A-C) and in the section 2.2 (D, E). The figure may be better to separate as Figure 2 and 3.

Author Response

Plaese, find the answers with additional graphics in the attached PDF document.

Round 2

Reviewer 2 Report

The paper is adequately revised. I recommend to accept in present form.